# Intrapartum Quality of Care among Healthy Women: A Population-Based Cohort Study in an Italian Region

**DOI:** 10.3390/ijerph21050629

**Published:** 2024-05-15

**Authors:** Simona Fumagalli, Antonella Nespoli, Maria Panzeri, Edda Pellegrini, Michele Ercolanoni, Paul Stefan Vrabie, Olivia Leoni, Anna Locatelli

**Affiliations:** 1School of Medicine and Surgery, University of Milano Bicocca, 20900 Monza, Italy; simona.fumagalli@unimib.it (S.F.); antonella.nespoli@unimib.it (A.N.); anna.locatelli@unimib.it (A.L.); 2Department of Obstetrics, Foundation IRCCS San Gerardo dei Tintori, 20900 Monza, Italy; 3Maternal and Child Committee, Lombardy Region, 20124 Milan, Italy; edda_pellegrini@regione.lombardia.it; 4ARIA, 20124 Milan, Italy; michele.ercolanoni@ariaspa.it (M.E.); paulstefan.vrabie@ext.ariaspa.it (P.S.V.); 5Welfare Department, Epidemiologic Observatory, Lombardy Region, 20124 Milan, Italy; olivia_leoni@regione.lombardia.it

**Keywords:** midwifery care, birth volume, perinatal outcomes, best practice, intrapartum care, quality of care

## Abstract

Although the quality of care during childbirth is a maternity service’s goal, less is known about the impact of the birth setting dimension on provision of care, defined as evidence-based intrapartum midwifery practices. This study’s aim was to investigate the impact of hospital birth volume (≥1000 vs. <1000 births/year) on intrapartum midwifery care and perinatal outcomes. We conducted a population-based cohort study on healthy pregnant women who gave birth between 2018 and 2022 in Lombardy, Italy. A total of 145,224 (41.14%) women were selected from nationally linked databases. To achieve the primary aim, log-binomial regression models were constructed. More than 70% of healthy pregnant women gave birth in hospitals (≥1000 births/year) where there was lower use of nonpharmacological coping strategies, higher likelihood of epidural analgesia, episiotomy, birth companion’s presence at birth, skin-to-skin contact, and first breastfeeding within 1 h (*p*-value < 0.001). Midwives attended almost all the births regardless of birth volume (98.80%), while gynecologists and pediatricians were more frequently present in smaller hospitals. There were no significant differences in perinatal outcomes. Our findings highlighted the impact of the birth setting dimension on the provision of care to healthy pregnant women.

## 1. Introduction

Worldwide, the aim of healthcare maternity services has expanded beyond the mother and child’s survival, focusing on quality of care and maternal experience [1,2,3]. Quality of care is characterized by “safe, effective, timely, efficient, equitable and people-centered” care. According to the WHO framework “provision of care”, “experience of care”, and “availability of physical and competent and motivated human resources” represent the three domains for improving the Quality of Maternal and Newborn Care (QMNC) [4]. The domain “provision of care” includes the evidence-based midwifery practice during routine and emergency care, information, and referral systems [4,5]. Best practices, including promoting nonpharmacological coping strategies, guaranteeing the presence of a birth companion, skin-to-skin contact, and promoting first breastfeeding, and intrapartum interventions, such as the use of epidural analgesia or episiotomy, are determinant factors in short- and long-term outcomes for mothers and babies [2,3,6,7,8,9]. Regarding the domain “availability of competent and motivated human resources” [4], evidence-based care should be provided by skilled healthcare providers (HCPs) [2,3,10,11] in a well-functioning healthcare system [2,3,12,13,14,15]. During birth, a midwife’s presence is associated with less intrapartum intervention, such as oxytocin augmentation and episiotomy, a higher rate of spontaneous vaginal birth, and a reduction in maternal adverse outcomes, such as uterine atony or hysterectomy and postpartum wound infection or sepsis [11,16,17].

Besides the QMNC domains [4], maternal and neonatal outcomes could also be influenced by organizational aspects [18,19], including the model of care [5,20,21] and birth setting characteristics. Place of birth, geographical location (rural versus urban), and birth volume [14,20,22] are the most studied elements of settings regarding their relationship with perinatal outcomes.

Considering low- and high-obstetric-risk pregnant women, some authors report a positive effect of high-volume hospitals in improving maternal outcomes [23,24], with a lower intervention intrapartum rate [14,22] and a higher spontaneous vaginal birth rate [25]; on the contrary, some others describe increased adverse maternal outcomes in high-volume hospitals [16,26]. Similarly, considering only healthy pregnant women, the literature is inconsistent about the relationship between the mode of birth and the hospital volume [15,25]. Several studies report a positive role of midwifery-led units [14,20,27] and urban hospitals [22,28] in achieving better perinatal outcomes, but the literature is not consistent about the relationship between birth volume and perinatal outcomes [15,18,22].

Studies including high-risk newborns reported lower perinatal mortality and morbidity in hospitals with high birth volume than those with medium-low birth volume [29,30]. However, a systematic literature review including low-risk newborns showed an inconsistent association between all outcomes considered and birth volumes [31].

Only a few papers investigated the effect of birth volume on the provision of care in terms of the use of evidence-based midwifery practice in healthy pregnant women. Jolles et al. [15] showed an inverse relationship between hospital dimension and intrapartum interventions, with a higher risk of induction in low-risk women who gave birth in higher-volume birth centers. Overall, within healthy pregnant women, the existing knowledge is inconsistent about the relation between birth volume, midwifery care, and their effect on maternal and perinatal outcomes.

This study aimed to investigate the impact of hospital birth volume (hospitals with more than or equal to 1000 births per year versus hospitals with fewer than 1000 births per year) on intrapartum midwifery care and maternal and neonatal outcomes within healthy women.

## 2. Materials and Methods

We conducted a population-based cohort study between 2018 and 2022 in Lombardy, a northern Italian region that accounts for approximately 16% of the country’s population (almost 10 million inhabitants).

### 2.1. Study Source

In Italy, a Certificate of Delivery Assistance (called CeDAP) is completed by the midwife who attends each birth and is consistently managed in a national database. This database collects data about (1) sociodemographic and obstetric women’s characteristics; (2) antenatal, intrapartum, and postnatal care; (3) maternal and perinatal outcomes. All Italian regions continuously fill out this database, allowing the evaluation and the improvement of the quality of the maternity services [32]. Moreover, every Italian region could refer to a Healthcare Utilization database, which offers information about services provided to all the citizen beneficiaries of the National Health Services (NHS), including (i) demographic and administrative data for all beneficiaries of the Regional Health Service (approximately coinciding with the entire resident population), considering residence municipalities; (ii) the hospital discharges registry, which reports all diagnoses released from public or private hospitals; and (iii) specialist visits and diagnostic exams registry. This database is completed both automatically (information i) and by the doctors who are responsible for the woman’s discharge (information ii and iii); the information from the Healthcare Utilization database is used for the refund of healthcare costs after the regional check of appropriateness.

This study’s source was composed of the national database of CeDAP, which is linked to Healthcare Utilization through a unique and personal identification code that is automatically converted into an anonymous code to maintain privacy. This procedure allowed the researchers to define the complete pathway of childbearing women enrolled in the study.

### 2.2. Setting

The Italian National Health Service guarantees all citizens equal and free access to healthcare services, including free maternity care during childbirth. The State-Regions Agreement of 16 December 2010 suggested centralizing perinatal care in hospital maternity units with at least 1000 births/year with a progressing reduction in smaller ones. While women with high-risk pregnancies are recommended to give birth in hospitals with specific organizational, structural, and technical standards, healthy women can choose their place of birth. Nationally, there are mostly obstetric-led units and few midwifery-led units [33,34,35], leading women to choose mostly between obstetric units classified according to birth volume (<1000 or ≥1000 births/year) and organizational standards [36].

In 2022, there were 387,934 births nationwide, almost exclusively in hospital settings; of these, 66,918 births occurred in Lombardy (17.25%) and more than 70% in hospitals with more than 1000 births per year [32].

Figure 1 shows the hospitals’ distribution within birth volume categories (hospitals ≥1000 births/year and hospitals <1000 births/year) in the Lombardy region in 2022. At the end of the study period (2022), there were 55 working obstetric units, and 24 of them (43.64%) had more than 1000 births per year, while 7 of them (12.73%) had more than 2500 births per year. Within smaller hospitals (n = 31), 11 had fewer than 500 births per year. Information on hospital birth volumes for each maternity hospital was obtained by the average of births occurring between 2018 and 2022.

### 2.3. Participants

Births that occurred in Lombardy between 1 January 2018 and 31 December 2022 were selected using the CeDAP database of Lombardy. Inclusion criteria were “healthy pregnancy”, according to two time-related criteria: (1) two years before the last menstrual period (absence of stillbirth, congenital malformation, any severe complication of pregnancy); (2) current pregnancy (maternal age 17–44 years, Body Mass Index 18–35, spontaneous conception, single pregnancy, gestational age 37–42 gestational weeks, no hospital admission during pregnancy or pregnancy complication, no congenital malformation, normal fetal growth, fetus alive). Exclusion criteria included breach or malpresentation at admission, inductions of labor or planned or prelabor, and Caesarean section, according to the regional definition of healthy pregnant women who are admitted to hospitals [37].

### 2.4. Variables

Sociodemographic information, such as the mother’s age, birthplace, education, and occupation, was analyzed. Concerning women’s obstetric history, parity was considered. To evaluate intrapartum midwifery care, information about best practices, intrapartum interventions, and healthcare professionals (HCPs) present at birth was collected for the entire sample. The best practices considered were the use of nonpharmacological coping strategies, the presence of a birth companion, skin-to-skin contact after birth, and first breastfeeding within 1 h after birth. Regarding intrapartum interventions, data about augmentation with amniotomy, the use of epidural analgesia or other pharmacological analgesia, and episiotomy were considered. The presence of a midwife, gynecologist, and pediatrician at birth was gathered through three binary categorical variables (presence/absence of midwife, presence/absence of pediatrician, and presence/absence of gynecologist). To investigate maternal and neonatal outcomes, we analyzed modes of birth through three binary categorical variables (presence/absence spontaneous vaginal birth, presence/absence vacuum assisted birth, and presence/absence Caesarean section), perineal integrity, physiologic blood loss (defined as <1000 mL), poor neonatal adaptation at birth (defined as Apgar score < 7 at 5 min), and transfer to another hospital. To reach the study’s aim, hospitals were classified into birth volume categories: high-volume birth hospitals (HV hospitals) were defined as hospitals with more than or equal to 1000 births per year, and low-volume birth hospitals (LV hospitals) were defined as hospitals with fewer than 1000 births per year.

### 2.5. Statistical Methods

The characteristics of the entire sample and the ones within the birth volume hospital categories were described using frequencies and percentages for categorical variables and using summary indicators [mean and standard deviation (SD)] for continuous variables. Distribution differences within birth volume hospital categories were tested using the Chi-square test (for categorical variables) and *t*-test (for continuous variables).

The association between maternity indicators and birth volume hospital categories was estimated using the log-binomial regression models, adjusted for the confounders of the parity, which mostly affected intrapartum interventions and perinatal outcomes: they allowed us to define the adjusted odds ratios (aORs) and their corresponding 95% confidence interval (95% CI). We decided to not control for the availability of resources because they are standardized according to Italian law [36], which clearly states the organizational (including healthcare professional availability), structural, and technical standards requested in all hospitals and that should be guaranteed for accreditation.

All tests performed were two-sided, and a *p*-value ≤ 0.05 was considered statistically significant.

## 3. Results

There were 352,969 women who gave birth between 2018 and 2022 in Lombardy, and 63.22% (n = 223,167) were healthy pregnant women who met the inclusion criteria. Of them, 77,943 (34.92%) were excluded for breech or malpresentation at admission and/or for induction and planned or prelabor CS. This study’s cohort included a total of 145,224 women, which represented 41.14% of the entire population of pregnant women in Lombardy from 2018 to 2022. (Figure 2). Among these births, 71.11% (n = 103,274) occurred in HV hospitals and 28.89% (n = 41,950) in LV hospitals. A total of 39,955 women gave birth in hospitals with more than 2500 births/year.

### 3.1. Sample’s Characteristics

The sociodemographic and obstetric characteristics of women are described in Table 1. The average age of mothers was 32.14 years, with a different distribution according to birth volume categories’ hospitals (*p*-value < 0.001). Women aged 16–25 gave birth more frequently in LV hospitals (14.62%). Women aged 36–45 gave birth more often in HV hospitals (28.03%). Women who gave birth in HV hospitals were more frequently Italian-born (*p*-value < 0.001), and they had a higher education level, such as a bachelor’s degree (*p*-value 0.004) or a master’s degree (*p*-value < 0.001). Furthermore, they were more likely to be employed (*p*-value < 0.001). Primiparous women (37.95%, n = 55,114) gave birth more frequently in HV hospitals (39.55% versus 34.02%, *p*-value < 0.001).

### 3.2. Impact of Hospital’s Birth Volume Categories on Intrapartum Midwifery Care

The distribution within the hospital’s birth volume categories of evidence-based intrapartum midwifery care, including best practices, intrapartum interventions, and HCPs present at birth, is reported in Table 2. Best practices were present in large percentages of all births. The presence of a birth companion was the most guaranteed (84.43%), followed by first breastfeeding (77.97%) and nonpharmacological coping strategies (75.47%), while skin-to-skin contact was ensured for 70.74% of women at birth. All practices were significantly differently distributed (*p*-value < 0.001) according to birth volume. The use of nonpharmacological coping strategies was offered more in LV hospitals (aOR 0.802; *p*-value < 0.001), while all other best practices considered were adopted more in HV hospitals (*p*-value < 0.001). Regarding intrapartum interventions, 25.49% of women received epidural analgesia, and 16.12% had an episiotomy. Women giving birth in HV hospitals had a higher recourse to epidural analgesia (aOR 2.263; *p*-value < 0.001) and episiotomy (aOR 1.077; *p*-value < 0.001). Considering the HCPs present at birth, midwives attended almost all the births (98.80%; n = 143,485), while the presence of gynecologists and pediatricians was, respectively, 71.73% (n = 104,174) and 43.54% (n = 63,231). The presence of gynecologists and pediatricians was more frequent in LV hospitals (aOR 0.34; *p*-value < 0.001; aOR 0.435; *p*-value < 0.001, respectively), where we observed a lower rate of best practices offered at birth, such as skin-to-skin contact and first breastfeeding within 1 h.

### 3.3. Impact of Hospital’s Birth Volume Categories on Maternal and Neonatal Outcomes

Maternal and neonatal outcomes according to the hospital’s birth volume categories are described in Table 3. A total of 126,747 women (87.30%) had a spontaneous vaginal birth, while 7268 (5.10%) had a vacuum-assisted birth, and 11,025 (7.60%) had a Caesarean section. Vacuum-assisted vaginal deliveries were less likely to occur in HV hospitals (aOR 0.724; *p*-value < 0.001), while there were no significant differences for the other modes of births. Perineal integrity was reported in 42.84% (n = 62,214) of births, occurring less frequently in HV hospitals (aOR 0.89; *p*-value < 0.001). Similarly, physiological blood loss was less prevalent in these hospitals (aOR 0.911; *p*-value 0.003). Regarding neonatal outcomes, a low Apgar score at 5 min (<7) was described in 597 newborns (0.41%), with no significant difference between hospital birth volume categories (*p*-value 0.113).

## 4. Discussion

Our population-based study reported the effect of hospital birth volume on the provision of intrapartum care and maternal and neonatal outcomes in a large sample of low-risk laboring women, a field which, according to our best knowledge, has never been investigated. Our study showed that more than 70% of healthy pregnant women gave birth in HV hospitals, which usually offer broader and more specialized services, potentially increasing the risk of medicalization during birth. The medicalization of childbirth has historically transformed birth into a dehumanized and mechanistic process that requires high-technology techniques and medical surveillance [38]. High medicalization has been described in the Italian context, and it could influence Italian mothers’ attitudes toward birth, leading them to perceive birth as a risky event [32,33,39]. The fear of adverse events during childbearing has led Italian women to seek highly specialized care, which is most often perceived as guaranteed in larger hospitals [40]. Indeed, safety is the most cited reason as to why women choose the place of birth, defining an important aspect of the care that women are looking for [38,40,41,42]. In addition, larger hospitals are also more likely to offer some maternity services, such as epidural analgesia or neonatal intensive care, reassuring them when they think of the supposed “risks” with birth [40,41]. Unfortunately, in Italy, there is still a lack of midwifery units, and only a few women could have access to them. This unavoidable supply gap influences women’s decision making and childbirth culture. The literature strongly suggests that healthy pregnant women safely choose midwifery units when available with lower intrapartum intervention without a negative impact on perinatal outcomes [13,38]. Policymakers should take these elements into account when planning maternity services to ensure a positive birth experience for all women beyond safety. Moreover, policymakers should consider that women who could benefit from midwifery unit care represent a large part of the entire population, totaling 41% of our population.

Our results show that healthy women who give birth in HV hospitals could increase their risk of intrapartum intervention exposure. The higher likelihood of epidural analgesia and episiotomy in HV hospitals could be explained by the higher risk of staff shortages due to simultaneous births or complicated situations, which could decrease the probability of ensuring one-to-one midwifery care; moreover, the higher exposure to complicated births in these settings can cause the overtreatment of healthy women [16,43]. Furthermore, midwives could probably experience more time pressure and work overload due to simultaneous births, resulting in a higher rate of intervention, like vacuum-assisted birth, and an increase in pathological blood loss [43,44,45,46]. Another result from this study confirms the hypothesized relation between the use of epidural analgesia and the different offerings of one-to-one midwifery care in HV and LV hospitals: nonpharmacological coping strategies were less frequently offered in HV hospitals, probably due to the lower offerings of one-to-one care [16,47]. The higher use of epidural analgesia could increase the rate of operative vaginal birth and, consequently, the rate of episiotomy, as confirmed by our data [48]. Interestingly we observed, within HV hospitals, higher rates of best practices related to the postpartum period, including skin-to-skin contact and breastfeeding within the first hour. This could be associated with a lower presence of gynecologists and pediatricians at birth. According to the literature, a motivated and competent staff is a determinant enabler for these best practices, and midwives promote and protect these practices more often than other HCPs [44,45]. Mother–newborn separation, in fact, appears to be more related to common practice than medical reasons [49,50]. In larger hospitals, gynecologists and pediatricians had a higher likelihood of being involved in caring for high-risk mothers and newborns and a lower likelihood of being involved in low-risk births. We observed that midwives were present at almost all births, without differences between birth volume categories, as recommended by the literature [21,51]. Interestingly, in Italy, the presence of a midwife during labor is ensured in most maternity services [35], while midwife-led care during the childbearing pathway is offered only to a small population of women. Therefore, during pregnancy, birth, and the postnatal period, different HCPs are involved in the process of care. This fragmentation of care could result in a loss of the holistic vision of normal birth and substandard care, especially during the postpartum period [52,53,54,55].

Our results suggest that maternity care policies should support healthy women in choosing LV hospitals, where their exposure to best practices is higher and the risk of intrapartum interventions is lower, without differences in modes of birth and perinatal outcomes. HCPs involved in maternity care should inform women of these results, supporting them in the decision-making process of birth setting, and policymakers should ensure the equitable access of all women to the right setting for their births. Moreover, in LV hospitals, an effort to reduce the involvement of gynecologists and pediatricians in normal births should be promoted. These findings could increase policymakers’ awareness of the educational and organizational needs of the regional territory; including tailored interventions promoting best practices and evidence-based midwifery care that should be implemented in LV and HV hospitals. Policymakers should improve the organizational aspects that could hamper HCPs in ensuring best practice and evidence-based midwifery care.

Furthermore, these findings could be considered by policy decision makers to implement different places of birth, such as outside of hospitals or inside hospitals but with independent organizational and structural functions; this implementation would be coherent with the regional law that suggested independent pathways for healthy pregnant women [37].

The strength of a population-based study is the ability to identify a specific health condition, avoiding the risk of selection bias; the size and comprehensive nature of our database allows us to describe a real picture of the provision of intrapartum midwifery care and the perinatal outcomes of healthy women within different birth volume hospital categories [56].

The retrospective nature of this study represents the main limitation of our study. Another limitation of this study is the lack of data on important factors of midwifery care, such as the model of care during pregnancy and one-to-one care during labor, and on perinatal outcomes, including maternal birth satisfaction, which may contribute to some unavoidable sources of uncertainty. Further research should implement our results in exploring other determinant aspects, including the model of care during pregnancy, the preference of the women, one-to-one care, and maternal birth satisfaction.

In addition, our study did not consider the effect of the COVID-19 pandemic on the provision of care, which was also highlighted in the literature in a local context [57,58,59,60].

## 5. Conclusions

Our population-based cohort study on birth within healthy pregnant women showed that hospital volume had an impact on the provision of care, including best practices and intrapartum interventions, and the presence of HCPs at birth. All these organizational aspects should be considered by healthcare providers and policy decision makers in developing organizational, functional, and economic strategies to promote maternal and neonatal health and reduce medicalization. Finally, to ensure a high quality of intrapartum care, a systematic measure of quality standards needs to be improved at local and regional levels.

## Figures and Tables

**Figure 1 ijerph-21-00629-f001:**
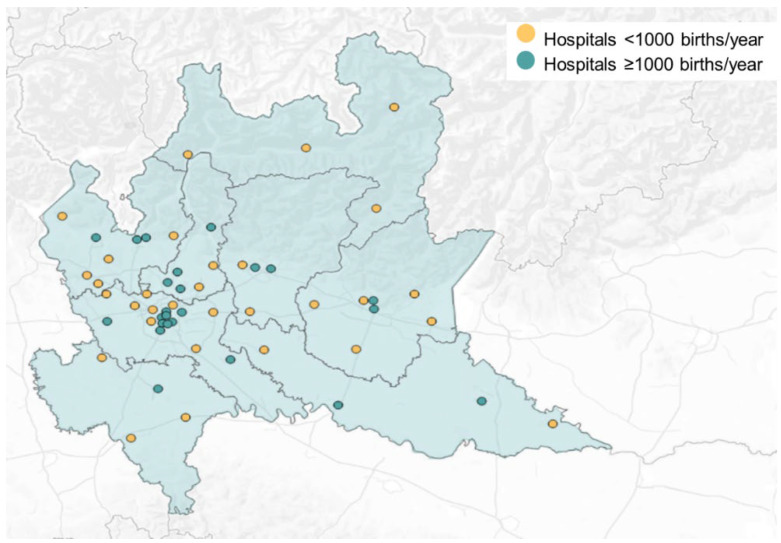
Hospital distribution within the hospital birth volume categories in Lombardy in 2022.

**Figure 2 ijerph-21-00629-f002:**
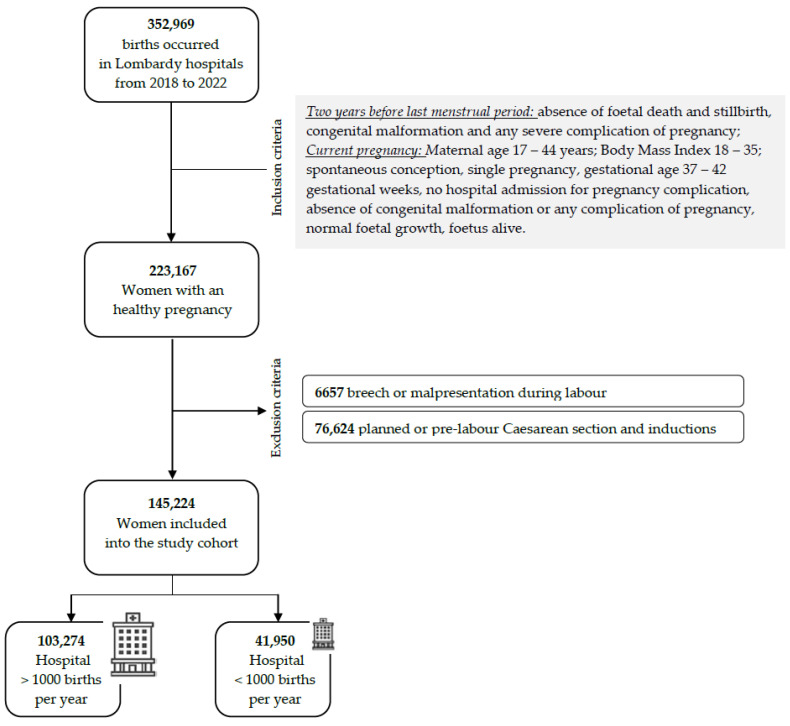
Flowchart of inclusion and exclusion criteria.

**Table 1 ijerph-21-00629-t001:** Distribution of sociodemographic characteristics and parity of the entire sample and within the hospital birth volume categories.

	Overall		LV Hospitals		HV Hospitals		*p*-Value
	*n*		*n*	*%*	*n*	*%*	
Number of healthy pregnancies	145,224		41,950	28.89	103,274	71.11	
	mean	SD	mean	SD	mean	SD	
Age	32.14	5.09	31.34	5.29	32.44	4.97	
	n	%	n	%	n	%	
16–25	15,732	10.83	6131	14.62	9601	9.30	<0.0001
26–35	90,613	62.40	26,140	62.31	64,473	62.43	
36–45	38,371	26.42	9421	22.46	28950	28.03	
Birth place	n	%	n	%	n	%	
Italian born	97,239	66.96	25,189	60.05	72,050	69.77	<0.0001
Foreign born	47,985	33.04	16,761	39.95	31,224	30.23	
Education	n	%	n	%	n	%	
Primary education	28,919	19.91	11,908	28.39	17,011	16.47	<0.0001
High school	60,618	41.74	19,200	45.77	41,418	40.10	0.250
Bachelor	7589	5.23	2064	4.92	5525	5.35	0.004
Master	48,034	33.08	8763	20.89	39,271	38.03	<0.0001
Missing	64	0.04	15	0.04	49	0.05	0.147
Occupation	n	%	n	%	n	%	
Employed	96,861	66.70	23,427	55.85	73,434	71.11	<0.0001
Housewife	33,010	22.73	14,127	33.68	18,883	18.28	<0.0001
Student	1208	0.83	341	0.81	867	0.84	<0.0001
Looking for first employment	147	0.10	51	0.12	96	0.09	0.102
Unemployed	13,748	9.47	3929	9.37	9819	9.51	0.653
Others	155	0.11	63	0.15	92	0.09	0.001
Missing	95	0.07	12	0.03	83	0.08	<0.0001
Parity							
Primiparous	55,114	37.95	14,271	34.02	40,843	39.55	<0.0001

**Table 2 ijerph-21-00629-t002:** Distribution of intrapartum midwifery practices in the entire sample and within the hospital birth volume categories and the impact of hospital’s birth volume categories on intrapartum midwifery practices.

Best Practices	Overall (n = 145,224)		LV Hospitals		HV Hospitals		aOR ^+^	95% CI ^++^	*p*-Value
	*n*	%	*n*	%	*n*	%			
Nonpharmacological coping strategies	109,601	75.47	33,955	80.94	75,646	73.25	0.802	[0.775; 0.83]	<0.0001
Presence of birth companion	122,618	84.43	33,326	79.44	89,292	86.46	1.530	[1.481; 1.581]	<0.0001
Skin to skin *	102,728	70.74	26,971	64.29	75,757	73.36	1.393	[1.357; 1.429]	<0.0001
First breastfeeding within 1 h	113,224	77.97	31,537	75.18	81,687	79.10	1.218	[1.183; 1.254]	<0.0001
Intrapartum interventions									
Augmentation with amniotomy	4100	2.82	939	2.24	3161	3.06	0.995	[0.705; 1.406]	0.963
Epidural analgesia	37,016	25.49	5917	14.10	31,099	30.11	2.263	[2.197; 2.330]	<0.0001
Nonepidural analgesia	7257	5.00	3229	7.70	4298	4.16	0.573	[0.544; 0.605]	<0.0001
Episiotomy *	23,411	16.12	6406	15.27	17,005	16.47	1.077	[1.044; 1.110]	<0.0001
Healthcare providers present at birth									
Midwife	143,485	98.80	41,393	98.67	102,092	98.86	0.956	[0.861; 1061]	0.397
Gynecologist **	104,174	71.73	36,130	86.13	68,044	65.89	0.34	[0.33; 0.349]	<0.0001
Pediatrician **	63,231	43.54	23,939	57.07	39,292	38.05	0.446	[0.435; 0.457]	<0.0001

* Within spontaneous vaginal births and vacuum-assisted births. ** Within spontaneous vaginal births. ^+^ aOR: odds ratio adjusted for parity. ^++^ 95% CI: 95% confidence interval.

**Table 3 ijerph-21-00629-t003:** Distribution of maternal and neonatal outcomes in the sample and within the hospital’s birth volume categories and the impact of hospital’s birth volume categories on maternal and neonatal outcomes.

Maternal and Neonatal Outcomes	Overall (n = 145,224)		LV Hospitals		HV Hospitals		aOR ^+^	95% IC ^++^	*p*-Value
Spontaneous vaginal birth	126,747	87.30	36,357	86.67	90,390	87.52	0.771	[0.572; 1.039]	0.081
Vacuum-assisted birth	7268	5.10	1885	4.49	5383	5.21	0.724	[0.535; 0.981]	0.042
Caesarean section	11,025	7.60	3680	8.77	7345	7.11	0.754	[0.558; 1.019]	0.068
Perineal integrity *	62,214	42.84	18,914	45.09	43,300	41.93	0.89	[0.869; 0.912]	<0.0001
Blood loss <1000 mL	140,188	96.53	40,582	96.74	99,606	96.45	0.911	[0.856; 0.97]	0.003
Low Apgar score at 5 min (<7)	597	0.41	189	0.45	408	0.40	0.872	[0.736; 1.033]	0.113

* Within spontaneous vaginal births; ^+^ aOR: odds ratio adjusted for parity; ^++^ 95% CI: 95% confidence interval.

## Data Availability

Restrictions apply to the availability of these data. Data were obtained from the Lombardy Region and they are available with the permission of the Lombardy Region.

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
