# Peer review of "Intrapartum Quality of Care among Healthy Women: A Population-Based Cohort Study in an Italian Region"

_ijerph, 2024, doi:10.3390/ijerph21050629_

Round 1

Reviewer 1 Report

Comments and Suggestions for Authors

Thank you very much for your manuscript, and congratulations on your research and for sharing it. Below, I present the evaluation in case you consider implementing the improvement points for your article.

The bibliography needs to be reviewed again and updated with more current information.

The introduction provides important geographical and demographic context to understand the scale and scope of the present study, as well as its assessment, relevance, and generalizability of findings within the Italian context and possibly broader. Overall, this introduction offers a comprehensive literature review and a solid justification for the proposed cohort study, although it could improve in its clearer and more concise structure, focused on the study objectives and avoiding redundancy.

In the methodology section, the database provides a robust foundation for gathering information on sociodemographic characteristics, obstetric history, intrapartum care, and maternal and neonatal outcomes.

The classification of hospitals based on birth volume and the description provide a clear understanding of the study environment.

The variables analyzed are relevant to the study objectives, and the data collection provides detailed information on the study's purpose.

The study presents some limitations that should be considered when interpreting the results. Firstly, its retrospective nature implies certain restrictions on the availability and accuracy of data, which could introduce information biases and hinder the capture of all relevant factors. Additionally, the lack of detailed data on certain aspects of obstetric care and outcomes may have limited the study's ability to fully address all relevant variables. Moreover, while some confounding factors were adjusted for in the analysis, there may be others that were not controlled for, such as the severity of cases or the availability of resources in different types of hospitals, which could have influenced the results.

To publish this article for the scientific community would be an excellent way to share these important findings. Congratulations on your work!!!!

Reviewer 2 Report

Comments and Suggestions for Authors

I commend your work on the "Intrapartum Quality of Care among Healthy Women: A Population-Based Cohort Study in an Italian Region" manuscript. Overall, the manuscript provides valuable insights into the implications of these findings for maternity care practices and policies. The authors contextualise their results within the existing literature and provide plausible explanations for the observed differences in care practices and outcomes between hospitals with high and low birth volumes. However, there are several areas where the manuscript could be improved, and the following recommendations aim to enhance the quality. They are not meant as a personal critique of the effort invested in the work.

General Comments:

-        Overall, the manuscript is well-written and structured. However, there are instances where complex sentences could be simplified for clarity, particularly in the methods section. Ensure consistency in terminology and definitions throughout the manuscript to avoid confusion.

Specific Comments:

Introduction:

-        While the introduction identifies the research aim to investigate the impact of hospital birth volume on intrapartum midwifery care and maternal and neonatal outcomes within healthy women, it would benefit from a clearer statement of the specific gap in the literature that this study seeks to address. What existing knowledge or evidence is lacking in this area?

-        The introduction discusses the inconsistent findings in the literature regarding the relationship between hospital volume and perinatal outcomes. Still, it could be strengthened by providing more context on why hospital volume is an important factor to consider. Why might higher or lower birth volume hospitals influence the quality of care and outcomes?

-        The introduction briefly touches on various studies on the relationship between hospital volume and perinatal outcomes, but the discussion feels somewhat disjointed. Consider organising the discussion more cohesively by grouping studies based on their findings or methodologies, which would help readers better understand the current state of the literature.

-        The introduction mentions that the study specifically focuses on healthy women, but providing a rationale for this choice would be beneficial. Why is it important to examine the impact of hospital birth volume, specifically within this population? Are unique considerations or factors making this group particularly relevant to the research question?

Methodology:

-        The description of the data sources, including the CeDAP and Healthcare Utilisation databases, is informative. However, it might be helpful to briefly explain how these databases are maintained, who contributes data to them, and how data quality is ensured. This would give readers a better understanding of the reliability and validity of the data used in the study.

-        The inclusion and exclusion criteria for participant selection are clearly stated, which is essential for understanding the study population. However, it might be helpful to provide a rationale for each criterion, particularly for the exclusion criteria related to breech or malpresentation at admission, inductions of labour, and planned or pre-labor Cesarean section. Explaining why these criteria were chosen would enhance transparency and help readers understand the rationale behind the study design.

-        The section provides a detailed description of the variables analysed in the study, including sociodemographic information, obstetric history, intrapartum midwifery care practices, and maternal and neonatal outcomes. This comprehensive approach is commendable and ensures that key factors influencing the outcomes of interest are considered. However, it might be helpful to clarify how variables were measured or categorised, particularly for categorical variables such as mode of birth and presence of healthcare professionals at birth.

-        The statistical methods used for data analysis are appropriate for the study objectives and are clearly described. However, providing a brief rationale for choosing statistical tests, such as Chi-square tests for categorical variables and t-tests for continuous variables would be beneficial. Additionally, it might be helpful to explain why log-binomial regression models were chosen for estimating associations between maternity indicators and hospital birth volume categories, particularly in terms of their advantages over alternative regression models.

Results:

-        The authors effectively interpret the findings by discussing the impact of hospital birth volume categories on various aspects of intrapartum midwifery care and maternal and neonatal outcomes. They also provide adjusted odds ratios (aOR) to quantify the associations between hospital birth volume and these outcomes, enhancing the results' interpretability.

-        The authors discuss the differences observed between hospitals with high and low birth volumes regarding sociodemographic characteristics, intrapartum midwifery care practices, and maternal and neonatal outcomes. However, it would be beneficial to provide further interpretation of these differences and discuss their potential reasons. For example, why are certain best practices more common in hospitals with high birth volumes? Are there specific organisational or structural factors that contribute to these differences?

-        The authors adjust for confounders such as maternal age and parity when examining the association between hospital birth volume and outcomes, strengthening the findings' validity. However, it would be helpful to provide more information on the selection of these confounders and discuss any other factors that may have been considered in the analysis.

-        While the results highlight important findings related to the impact of hospital birth volume on intrapartum care and outcomes, it would be valuable to acknowledge any study limitations. For example, are there potential biases or sources of error that could have influenced the results? Discussing limitations would give readers a more comprehensive understanding of the study's scope and implications.

-        The authors briefly touch upon the clinical implications of their findings, such as the need to ensure equitable access to evidence-based midwifery care across different hospital settings. Expanding on the clinical implications and discussing potential strategies for improving care quality in high- and low-volume hospitals would add depth to the discussion section.

Discussion:

-        The authors effectively contextualise their findings within the existing literature, highlighting the unique contribution of their study to the field. However, it would be beneficial to explore further the implications of their findings about broader trends in maternity care practices and policies, both within Italy and internationally. How do their findings align with or diverge from global trends in maternity care, particularly regarding the balance between medicalisation and holistic, woman-centred care?

-        The authors offer plausible explanations for the differences in intrapartum care practices and outcomes between hospitals with high and low birth volumes. However, it would be valuable to delve deeper into the underlying mechanisms driving these differences. For example, what specific organisational or structural factors within high-volume hospitals may contribute to higher rates of interventions such as epidural analgesia and episiotomy? Exploring these mechanisms would provide greater insight into the drivers of variation in care quality across different hospital settings.

-        While the authors acknowledge potential explanations for their findings, such as staff shortages and workload pressures in high-volume hospitals, it would be prudent to consider alternative explanations or confounding factors that may have influenced the results. For example, are there differences in patient preferences or provider attitudes toward intervention that could contribute to the observed disparities in care practices?

-        The authors appropriately discuss the clinical implications of their findings, emphasising the importance of supporting healthy women in choosing low-volume hospitals where they may have a lower risk of intrapartum interventions and higher exposure to best practices. However, it would be helpful to provide more specific recommendations for policymakers and healthcare providers based on the study findings. For example, how might maternity care policies be redesigned to promote equitable access to evidence-based care across different hospital settings?

-        The authors acknowledge the limitations of their study, including its retrospective nature and the lack of data on certain factors that may influence care practices and outcomes. It would be valuable to discuss potential avenues for future research to address these limitations and further elucidate the complex relationship between hospital birth volume, care practices, and outcomes. Additionally, considering the potential impact of the COVID-19 pandemic on maternity care delivery, as suggested by the literature, could enrich the discussion and provide additional insights into the broader context of the study findings.

Comments on the Quality of English Language

Moderate editing of the English language is required.

Round 2

Reviewer 2 Report

Comments and Suggestions for Authors

Thank you for re-reading the manuscript "Intrapartum Quality of Care among Healthy Women: A Population-Based Cohort Study in an Italian Region". On the basis of the rereading and the comments made previously, I conclude that the authors have taken into account the suggested earlier comments and have significantly improved the quality of the manuscript. I have no further comments for the authors. Congratulations on your successful work!